# *Pediococcus pentosaceus* MIANGUAN2 Alleviates Influenza Virus Infection by Modulating Gut Microbiota and Enhancing Short-Chain Fatty Acid Production

**DOI:** 10.3390/nu16121923

**Published:** 2024-06-18

**Authors:** Yulu Chen, Liqiong Song, Mengshan Chen, Yuanming Huang, Zhihuan Wang, Zhihong Ren, Jianguo Xu

**Affiliations:** 1National Key Laboratory of Intelligent Tracking and Forecasting for Infectious Diseases, National Institute for Communicable Disease Control and Prevention, Beijing 102206, China; 2Institute of Public Health, Nankai University, Tianjin 300071, China

**Keywords:** *Pediococcus pentosaceus*, anti-influenza effect, cytokines, gut microbiota, short-chain fatty acids

## Abstract

Influenza, a severe respiratory disease caused by the influenza virus, has long been a prominent threat to human health. An increasing number of studies have demonstrated that oral administration with probiotics may increase the immune response to lung infection via the gut-lung axis leading to the alleviation of the pulmonary disease. In this study, we evaluated the effects of oral administration of *Pediococcus pentosaceus* MIANGUAN2 (MIANGUAN2) on influenza infection in a mouse model. Our results showed that oral administration of MIANGUAN2 significantly improved weight loss, lung index, and lung pathology, and decreased lung viral load of influenza-infected mice. Additionally, MIANGUAN2-treated mice showed significantly lower levels of TNF-α, IL-1β, IFN-γ, and IL-12p70 and higher production of IL-4 in the lung. In accordance with this, the transcriptome analysis of the lung indicated that MIANGUAN2-treated mice had reduced expression of inflammation markers, such as TNF, apoptosis, and the NF-Kappa B pathway. Furthermore, the administration of MIANGUAN2 restored the SCFAs profiles through regulating the gut microbiota. SCFA-producing bacteria, such as p_Firmicutes, f_Lachnospiraceae, and f_Ruminococcaceae, were enriched in the MIANGUAN2-treated group compared with PBS-treated group. Consistently, the concentrations of SCFAs in the MIANGUAN2 group were significantly higher than those in the PBS-treated group. In addition, the concentrations of SCFAs were positively correlated with SCFA-producing bacteria, such as *Ruminococcus*, while being negatively correlated with the virial titers and proinflammatory cytokines. In conclusion, this animal study suggests that *Pediococcus pentosaceus* MIANGUAN2 may alleviate the influenza infection by altering the gut microbiota composition and increasing the levels of gut microbiota-derived SCFAs.

## 1. Introduction

Influenza is a viral respiratory infection that affects all age groups and can cause high morbidity and mortality during pandemics [1]. Recent investigations show that about 650,000 global influenza-associated deaths occur every year [2], emphasizing the urgency of preventing influenza infection. Influenza vaccination at optimal timing is an effective intervention to protect susceptible populations [3]. However, due to the higher rates of annual mutations of the influenza virus, vaccines are difficult to match with epidemic strains, which poses a great challenge to vaccination efficacy. In addition, although certain antiviral drugs serve a significant prophylactic and therapeutic purpose to decrease influenza-associated morbidity [4], the continuous emergence of resistant viral strains can reduce their effectiveness for influenza treatment [5]. Hence, it is necessary to find alternative effective strategies to prevent and treat influenza infection. 

The gastrointestinal microbiome has been shown to play a significant role in host immunity [6]. Influenza infection can cause the loss of food intake [7,8,9] and inflammation or barrier dysfunction of the gut [10], thereby causing dysbiosis of healthy gut microbiota. Increasing evidence indicates that the disruption of the host gut microbiome can adversely influence anti-influenza viral immunity. Bradley KC et al. have found that the gut microbiota is critical for driving an interferon (IFN) signature in lung stroma cells; antibiotic treatment can disrupt the gut microbiota and the lung stromal IFN signature of mice and facilitate the early replication of the influenza virus in the lung, leading to severe influenza virus infection [11]. “Probiotics” are defined as live microorganisms that are beneficial to the host when taken in sufficient quantities [12]. Oral administration with probiotics may change the composition of gut microbiota to help combat lung disease though the gut-lung axis [13,14]. In recent years, studies have shown that oral administration with certain probiotics can significantly alleviate influenza infection through regulating gut microbiota. Treatment with novel immunobiotics, such as *Bacteroides* dorei, significantly improved the gut microbiota disorder and alleviated influenza infection in mice [15]. Qiang et al. have proved that the expansion of the gut population of endogenous *Bifidobacterium animalis* is one of the reasons for protecting the mice from lethal influenza infection [16].

Recent studies have extensively investigated the involvement of the gut-lung axis mechanism in the effect of gut microbiota on pulmonary infection [17,18]. Some evidence indicates that gut microbiota-derived metabolites are potential mediators of the gut-lung axis that play a pivotal role in protecting the host against respiratory virus infections. *N*-acetyl-d-glucosamine (GlcNAc) produced by gut microbiota (*Clostridium* sp., *Phocaeicola sartorii,* and *Akkermansia muciniphila*) protects the host against influenza infection by increasing the number and activity of NK cells [19]. Desaminotyrosine (DAT) is a gut microbiota-associated degradation product of flavonoids or amino acid metabolism that protects against influenza infection by upregulating type Ⅰ IFN signaling [20].The gut microbiota-derived (*Bifidobacterium pseudolongum* NjM1) acetate can enhance host anti-influenza response via enhancement of NLRP3-mediated (NOD-, LRR- and pyrin domain-containing 3) type Ⅰ IFN production [21]. Furthermore, oral administration of *Clostridium butyricum* can increase the production of 18-hydroxy eicosapentaenoic acid (18-HEPE) by gut microbiota, and the 18-HEPE can promote the production of IFN-λ through G protein-coupled receptor (GPR)120 and IFN regulatory factor (IRF)-1/-7 activation, thereby promoting resistance against respiratory virus infection in mice [22]. Oral administration of *L. paracasei* MI29 protects the host against influenza infection in a Gpr40/120-dependent manner by enhancing the expression of ISGs and changing the fatty acid profile of the gut, especially palmitic acids [23]. Thus, the administration of certain probiotics with the function of regulating the gut microbiota and metabolisms is a promising strategy for combating influenza infection.

*Pediococcus pentosaceus* is Gram-positive facultative anaerobic lactic acid bacteria, which has several beneficial properties, such as antimicrobial [24], detoxifying [25], lipid-lowering [26], anticarcinogenic [27], intestinal disease-alleviating [28,29], and immune-enhancing activities [30,31,32]. However, it is unknown whether oral administration with *P. pentosaceus* is effective against respiratory viral infections. Thus, we conducted the current study to test its potential effect in this regard.

Our results showed that treatment with MIANGUAN2 protected the mice from influenza virus infection, changed the gut microbiota composition, and increased the levels of SCFAs.

## 2. Materials and Methods

### 2.1. Bacteria and Virus

The strain of *Pediococcus pentosaceus* MIANGUAN2 used in this study was isolated from healthy human feces and preserved in the China General Microbiological Culture Collection Center (CGMCC) with the preservation number CGMCC No.29410. The genomic DNA of *P. pentosaceus* MIANGUAN2 was extracted using a Wizard^®^ Genomic DNA Purification Kit (Cat#A1125, Promega, Madison, WI, USA), and then sequenced on the Illumina PE150 platform by NOVOGENE (Beijing, China). For identification, the digital DNA-DNA hybridization [33] (dDDH, https://ggdc.dsmz.de/, accessed on 28 December 2023) and the average nucleotide identity [34] (ANI, https://www.ezbiocloud.net/tools/ani, accessed on 28 December 2023) were estimated between MIANGUAN2 and *P. pentosaceus* ATCC33316, and the results showed that dDDH and ANI were 97% and 99.64%, respectively. Thus, MIANGUAN2 can be identified as a strain of *Pediococcus pentosaceus*. The strain was cultured in de Man, Rogosa, and Sharpe (MRS) broth (Cat#CM0359B, Thermo Scientific™, Waltham, MA, USA) at 37 °C until OD_600nm_ = 1.0, and then the bacteria were centrifuged and washed three times with sterile phosphate-buffered saline (PBS) for subsequent application. Influenza virus A/Puerto Rico/8/34 mouse-lung-adaptive strain (H1N1, PR8) was propagated with MDCK and quantified as previously described [35].

### 2.2. Animals and Ethics Statement

The female C57BL/6J mice (13–15 g) were purchased from Beijing Vital River Laboratory Animal Technology Co., Ltd. (Beijing, China) [laboratory animal permit no. SYXK (Jing) 2022-0029], and they were housed in the Laboratory Animal Center of the Chinese Center for Disease Control and Prevention. The mice were housed 5 per cage under a 12-h light/dark cycle and had free access to water and food. All animal experiments were approved and conducted in accordance with the guidelines of the Institutional Animal Care and Use Committee in the Chinese Center for Disease Control and Prevention Laboratory Animal Center (Approval number: 2023-025).

### 2.3. Experimental Design and Sample Collection

The mice were randomly divided into the following three groups (*n* = 5 per group): control group (uninfected), virus group (PBS-treatment of infected mice), and MIANGUAN2 group (MIANGUAN2-treatment of infected mice). Except for the mice in the control group, all mice were anesthetized, followed by intranasal inoculation with 450 PFU PR8 on the infection day (Day 0). The mice in the virus and MIANGUAN2 groups were gavaged daily with 200 μL PBS and 5 × 10^9^ CFU MIANGUAN2, respectively, from day −5 (5 days prior to infection) to day 7 p.i. (7 days post infection). The body weight was recorded every day after infection. At day 7 p.i., the whole lungs were obtained to measure the lung index (lung index = lung weight/body weight × 100%), viral titer, inflammatory markers, and histopathology. The colon was collected to measure the length. The fecal samples and cecal content were collected for 16S rRNA gene sequencing analysis and metabolomics analysis, respectively. All samples were preserved at −80 °C.

### 2.4. Measurement of Lung Viral Titers

To determine viral titers, the whole lung was homogenized in 1 mL Trizol^TM^ reagent (Cat#15596018, Invitrogen, Carlsbad, CA, USA), followed by extraction of total RNA according to the manufacturer’s instructions. The total RNA was diluted to 1 μg/μL for real-time PCR analysis using a real-time PCR detection kit for influenza virus A (Cat#CN10-4C-100, Jiangsu Uninovo Biological Technology Co., Ltd., Jiangsu, China) on an Applied Biosystems 7500 real-time PCR system (Thermo Fisher Scientific, Waltham, MA, USA).

### 2.5. Quantitative Real-Time PCR

The extracted total RNA of lung tissues was reverse transcribed into cDNA using PrimeScript™ RT Master Mix (Cat#RR036A, Takara, Tokyo, Japan). The cDNA was used as a template for quantitative real-time PCR using TB Green^®^ Premix Ex Taq™ II (Cat#RR820A, Takara, Tokyo, Japan) on an ABI 7500 real-time PCR system. Relative expression levels of the target genes were calculated by the method of 2^−∆∆Ct^. The primer sequences for the detection of influenza virus titer were NP, 5′-GATTGGTGGAATTGGACGAT-3′, and 5′-AGAGCACC ATTCTCTCTATT-3′.

### 2.6. Histopathology Analysis of the Lung

Histopathology analysis was conducted by Wuhan Servicebio Technology Co., Ltd. (Wuhan, China). Lung tissues were collected and fixed in 4% paraformaldehyde solution for 24 h, followed by embedding in paraffin, then the tissues were stained with hematoxylin and eosin (H&E). Histopathological score was assessed as 0 to 4 by 3 pathologists in a double-blind setting based on the level of inflammatory cell infiltration, perivascular edema, bleeding, and the thickening of the alveolar walls [36].

### 2.7. Cytokine Measurement

The whole lung was homogenized in 1 mL UltraPure^TM^ distilled water (Cat#10977023, Invitrogen, Carlsbad, CA, USA) using a tissue grinder for 3 min, followed by centrifuging at 12,000 rpm for 5 min. The supernatant was obtained to quantify cytokines including TNF-α (Cat#88-7324, Invitrogen, Carlsbad, CA, USA), IL-1β (Cat#88-7013, Invitrogen, Carlsbad, CA, USA), IL-6 (Cat#88-7064, Invitrogen, Carlsbad, CA, USA), IFN-γ (Cat#88-7314, Invitrogen, Carlsbad, CA, USA), IL12p70 (Cat#88-7121, invitrogen, Carlsbad, CA, USA), IL-4 (Cat#88-7044, Invitrogen, Carlsbad, CA, USA), and IL-10 (Cat#88-7121, Invitrogen, Carlsbad, CA, USA) using mouse ELISA kits following the instructions of manufacturers.

### 2.8. Transcriptome Analysis of Lung Tissues

The lung tissues were collected on day 7 post-infection and were subjected to transcriptome analysis by Beijing Genomics Institute Co., Ltd. (Beijing, China). Specifically, the total RNA of lung tissue was extracted using Trizol according to the manufacturer’s instruction, and then the qualified RNA samples were used to construct a single-stranded circular DNA library. Next, the qualified DNA library was sequenced using the DNBSEQ platform [37]. The raw sequencing data were filtered with SOAPnuke to produce clean data. The subsequent analysis and data mining were performed on the Dr. Tom Multi-omics Data mining system (https://biosys.bgi.com, accessed on 4 January 2024). The “pheatmap” (v1.0.8) [38] was used to draw a heatmap of the differential genes in different lung tissues. The raw data were submitted to the NCBI SRA database (BioProject No: PRJNA1109985).

### 2.9. 16S rRNA Gene Sequencing Analysis of Gut Microbiota

The 16S rRNA gene sequencing analysis of fecal samples was conducted by Beijing Genomics Institute Co., Ltd. First, the total DNA of fecal samples (100–200 mg) was extracted, followed by a quality test. Then, the V3 and V4 hypervariable regions of 16S rDNA were amplified by PCR (5′-CCTACGGGNGGCWGCAG-3′ and 5′-GACTACHVGGGTATCTAATCC-3′). The qualified PCR products were sequenced on an Illumina Hiseq 2500 platform after quality inspection using an Agilent 2100 Bioanalyzer (Agilent, Santa Clara, CA, USA). After sequencing, the raw data were filtered to generate high-quality clean data, followed by obtaining the tags of high variable regions using FLASH (Fast Length Adjustment of Short Reads, v1.2.11) [39]. The tags were clustered into OTUs (operational taxonomic units) using USEARCH (v7.0.1090), and the representative OTU sequences were obtained using UPARSE according to a 97% threshold. Finally, the representative OTU sequences were aligned against the database for taxonomic annotation using the RDP Classifier (v2.2) software. After filtering the annotation results, the highly qualified OTU was used for post-analysis, including alpha diversity analysis (mothur, v.1.31.2) [40], beta diversity (QIIME, v1.80) [41], and linear discriminant-analysis effect size analysis (LEfSe). Furthermore, the abundance of the bacterial community was determined at different taxonomic levels. The original data can be downloaded from the NCBI SRA database (BioProject No: PRJNA1110029).

### 2.10. Metabolome Analysis of Cecal Content

The cecal content of mice was collected on day 7 post-infection to perform HM700 high-throughput targeted metabolomics analysis, which was conducted by Beijing Genomics Institute Co., Ltd. HM700 can achieve absolute quantification of more than 700 small molecules of metabolites in one time. In detail, 10 mg feces were mixed with 20 μL deionized water and 120 μL of 50% aqueous methanol. After grinding, the samples were centrifuged, and 30 μL of the supernatant was transferred to a 96-well plate, followed by the addition of 20 μL derivative reagents (200 mM 3-NPH in 75% aqueous methanol) and 20 μL EDC solution. Then, the 96-well plate was placed in a thermostatic oscillator at 40 ℃ for reaction for 60 min. After derivatization, the samples were centrifuged at 4 ℃ for 10 min, 30 μL of supernatant was transferred to a new 96-well plate, and 90 μL of 50% of aqueous methanol was added to dilute samples. Next, the samples were centrifuged at 4000× *g* and 4 ℃ for 5 min. The supernatant was used for UHPLC/MS analysis using a Waters ACQUITY UPLC I-Class Plus (Waters, Milford, MA, USA) with a QTRAP6500 plus (SCIEX, Framingham, MA, USA) [42]. Finally, the data were analyzed using Dr. Tom’s analysis platform (https://biosys.bgi.com/, accessed on 17 January 2024) via routine procedures.

### 2.11. Statistical Analysis

The GraphPad Prism 8.0.1 (GraphPad Inc., San Diego, CA, USA) and R software (version 4.3.3) were used to analyze and visualize the data, respectively. All data were expressed as means ± standard deviation (SD) (parametric statistics) or median with inter-quartile range (non-parametric statistics). The Mann-Whitney U tests and the two-tailed unpaired Student’s *t*-test were used to analyze the two groups’ non-normally and normally distributed data, respectively. For more than 3 group comparisons, the data were analyzed by one-way ANOVA with Tukey’s post hoc test (for normally distributed data) or Kruskal-Wallis’s test (for non-normally distributed data). The results were considered significantly different when the *p*-value was less than 0.05. For correlation analysis visualization, heatmaps and network diagrams were generated using the “pheatmap” [38] and “ggplot2” [43] packages, respectively. In addition, Spearman correlation analysis was conducted using the “psych” package [44].

## 3. Results

### 3.1. P. pentosaceus MIANGUAN2 Protects the Host against Influenza Virus Infection

To evaluate the effect of oral administration with MIANGUAN2 on resistance to influenza infection, female C57BL/6 mice were intranasally infected with the H1N1 influenza virus at day 0 and gavaged with PBS or MIANGUA2 from day −5 to day 7 (Figure 1A). The results showed less weight loss (Figure 1B), increased colon length (Figure 1C), and decreased lung index (Figure 1D) after influenza virus infection in MIANGUAN2-treated mice compared with PBS-treated mice. We also observed a significant decrease in lung viral titers in the lungs of mice treated with MIANGUAN2 compared with PBS-treated mice (Figure 1E,F). In addition, compared with PBS-treated mice, MIANGUAN2-treated mice exhibited less severe lung injury (Figure 1G) and lower lung pathological scores (Figure 1H), as indicated by lower levels of inflammatory cell infiltration, perivascular edema, bleeding, and thickening of the alveolar walls. Together, these results suggest that oral administration with the MIANGUAN2 strain may protect the host against H1N1 influenza infection.

### 3.2. P. pentosaceus MIANGUAN2 Regulates Cytokine Production and Suppresses the Expression of Multiple Inflammatory Signaling Pathways in the Lung

We examined the cytokine profiles of lung tissues in different groups of mice using ELISA. As shown in Figure 2A–G, the levels of the cytokines TNF-α (Figure 2A), IL-1β (Figure 2B), IL-6 (Figure 2C), IFN-γ (Figure 2D), and IL12p70 (Figure 2E) were significantly increased, and IL-4 (Figure 2F) level was decreased in the virus group compared with the control group. Oral administration with MIANGUAN2 significantly decreased the levels of proinflammatory cytokines, including TNF-α (virus vs. MIANGUAN2: 110.2 ± 12.9 vs. 81.5 ± 13.6), IL-1β (virus vs. MIANGUAN2: 313.1 ± 32.1 vs. 215.7 ± 68.1), IFN-γ (virus vs. MIANGUAN2: 3794.1 ± 496.2 vs. 2121.6 ± 788.7), and IL12p70 (virus vs. MIANGUAN2: 10.6 ± 3.3 vs. 4.0 ± 3.9), and increased the levels of cytokine IL-4 (virus vs. MIANGUAN2: 23.2 ± 2.7 vs. 31.5 ± 6.9) in the lung tissue of mice compared with those in the virus group. Specifically, the levels of TNF-α, IL-1β, IFN-γ, and IL12p70 in the MIANGUAN2 group were reduced by 26%, 31%, 44%, and 62% compared to the virus group, respectively. The levels of IL-10 were not significantly different among the three groups (Figure 2G).

To further explore the mechanisms by which MIANGUA2 reduces lung inflammation, we collected the lung tissues on day 7 post-infection to conduct the transcriptome analysis. The gene set enrichment analysis (GSEA) showed that the TNF (Figure 3A), apoptosis (Figure 3B), and NF-Kappa B signaling pathways (Figure 3C) were downregulated in the MIANGUAN2 group compared with the virus group. Next, we performed a hierarchical clustering analysis of the differentially expressed genes (DEGs) of these signaling pathways. Treatment with MIANGUAN2 downregulated the expression of *Tnfrsf1a* and *Tnfrsf1b* genes, which supports the results that MIANGUAN2 dramatically downregulated the TNF signaling pathway (Figure 3A, right panel) because Tnfrsf1a and Tnfrsf1b are the key receptors to activate the TNF downstream signaling pathway [45]. In the apoptosis pathway analysis, we observed downregulation of the apoptotic gene families, such as caspases *(Casp3*, *Casp7*, *Casp8*, *Casp12)* [46] and the B cell lymphoma (Bcl-2) family of genes (*Bak1*, *Bax*, *Bid*) [47] in the MIANGUAN2 group compared with the virus group (Figure 3B, right panel). Furthermore, oral administration with MIANGUAN2 also dampened NF-Kappa B signaling pathway by decreasing the expression of key genes *Traf2*, *Traf3*, and *RelB* [48] (Figure 3C, right panel). All these results suggest that oral MIANGUAN2 may alleviate lung inflammation through inhibiting multiple inflammatory signaling pathways in the lung.

### 3.3. P. pentosaceus MIANGUAN2 Alters the Gut Microbiota Composition of Influenza Infected-Mice

Previous studies have shown that the change in gut microbiota can modulate the host response to influenza infection [11,16]. To evaluate the effect of MIANGUAN2 on the gut microbiota structure of influenza-infected mice, we assessed the α-diversity by calculating the Chao1 and Shannon indexes among the three groups. The results showed that there was no significant difference in the Chao1 index among the three groups, whereas the Shannon index significantly increased in the MIANGUAN2 group compared with the virus group (Figure 4A), indicating that the MIANGUAN2 treatment restored the diversity of gut microbiota in influenza-infected mice. Next, in evaluating the β-diversity of gut microbiota using principal coordinates analysis (PCoA) based on UniFrac metrics, we found a distinct separation between the three groups (Figure 4B). To further identify the different taxa between the virus group and MIANGUAN2 groups, the LEfSe analysis was conducted. As shown in the cladogram, c_Clostridia, f_Lachnospiraceae, and f_Ruminococcaceae were enriched in the MIANGUAN2 group, while f_Enterobacteriaceae and f_Akkermansiaceae were predominant in the virus group (Figure 4C). Furthermore, the linear discriminant analysis score (LDA) (>4) showed a higher abundance of g_*Lachnospiraceae*_*NK4A136*, g_*Tyzzerella*, and g_*Pediococcus* in the MIANGUAN2 group, while there was a higher abundance of g_*Akkermansia*, g_*Escherichia*_*Shigella*, g_*Eubacterium*_*coprostanoligenes*, and g_*Faecalibaculum* in the virus group (Figure 4D). To further analyze the changes in gut microbiota at different taxonomic levels, we compared the relative abundance of the phyla, families, and genera among the three groups (Figure 4E–J). As shown in Figure 4E–G, the top 5 abundance of phyla as well as the top 10 abundance of families and genera were exhibited via the stacked bar plots of microbiota composition. In addition, the significance difference in microbiota was analyzed at phylum, family, and genus levels among the three groups, respectively. At the phylum level, the abundance of Firmicutes was higher in the MIANGUAN2 group compared with that in the virus group, whereas the relative abundances of Proteobacteria and Verrucomicrobiota were lower in the MIANGUAN2 group than those in the virus group (Figure 4H). At the family level, a higher abundance of Lachnospiraceae and Ruminococcaceae, as well as a lower abundance of Akkermansiaceae, Sutterellaceae, and Enterobacteriaceae, were observed in the MIANGUAN2 group, compared with those in the virus group (Figure 4I). At the genus level, the MIANGUAN2 group had higher abundances of the genera *Tyzzerella*, *Pediococcus*, and *Ruminococcus* and lower abundances of *Escherichia_Shigella*, *Eubacterium_nodatum*, *Parasutterella*, *Faecalibaculum*, *Eubacterium_coprostanoligenes*, *Enterobacter*, and *Akkermansia* than those in the virus group (Figure 4J).

### 3.4. P. pentosaceus MIANGUAN2 Increases the SCFA Levels in the Feces of Influenza-Infected Mice

Increasing evidence has proved that the change in gut microbiota structure affects the composition and content of microbial metabolites in the host, which plays a significant role in host health and disease state [49]. To explore the effect of MIANGUAN2 on the profile of gut microbial metabolism, the metabolic outputs among the three groups were measured by HM700 targeted metabolomics analyses. A total of 471 gut microbiota metabolites were identified and quantified, which were classified into 34 categories based on their final class. The top two most abundant metabolites were amino acids and peptides, as well as fatty acids (Figure 5A). The partial least squares-discriminant analysis (PLS-DA) showed significant separations between the three groups, indicating that the three groups have different gut microbiota metabolite profiles (Figure 5B). The results of the KEGG analysis of differential metabolites (*p* value < 0.05) are presented in Figure 5C. The carbohydrate digestion and absorption pathway, butanoate metabolism pathway, and lysine degradation metabolism pathway were upregulated, while the protein digestion and absorption pathway and fatty acid degradation pathway were downregulated in the MIANGUAN2 group compared with the virus group. The changes in these pathways are closely related to SCFAs metabolism. Next, we compared the levels of SCFAs among the three groups (Figure 5D). The levels of acetic acid, propionic acid, butyric acid, isobutyric acid, valeric acid, and isovaleric acid were significantly decreased in the virus group compared with those in the uninfected group. However, oral administration of MIANGUAN2 significantly restored the levels of acetic acid, propionic acid, butyric acid, isobutyric acid, and valeric acid (Figure 5E–I), while it did not significantly increase the levels of isovaleric acid (Figure 5J).

### 3.5. Correlations between SCFAs and the Gut Microbiota or Cytokine Profiles of Lung Tissues or Influenza Infection Phenotype

To elucidate the association of gut SCFAs production with the gut microbiota composition, we performed a Spearman’s correlation analysis. The results showed that the enriched genera in the MIANGUAN2 group, such as *Tyzzerella*, *Pediococcus*, and *Ruminococcus*, were positively corrected with SCFAs, whereas the decreased abundances of genera including *Escherichia_Shigella*, *Eubacterium_nodatum*, *Parasutterella*, *Faecalibaculum*, *Eubacterium_coprostanoligenes*, and *Akkermansia* in the MIANGUAN2 group were negatively correlated with SCFAs concentrations (Figure 6A). In addition, there was a significantly strong correlation between gut microbial taxa and SCFAs via plotting the network diagram based on the criteria of *p*-value < 0.05 (Figure 6B). Notably, among these SCFAs, the butyric acid had the strongest correlation with the differential genera.

The beneficial influence of SCFAs on health has been demonstrated in previous studies [50,51]. To clarify the association between the SCFAs and influenza infection phenotype of mice, a Spearman’s correlation analysis was conducted. As shown in the association heatmap (Figure 6C), the SCFAs were negatively associated with viral titer, lung index, and certain proinflammatory factors (TNF-α, IL-1β, IL-12, IFN-γ), and positively correlated with the colon length and anti-inflammatory cytokines (IL-4, IL-10). The network diagram exhibited significant associations between SCFAs and the clinical symptoms (Figure 6D). Our findings revealed that microbiota-derived SCFAs may play a significant role in combating influenza infection, especially in inhibiting pulmonary inflammation.

## 4. Discussion

An increasing number of studies have proved that certain probiotics can protect the host against various respiratory virus infections, including the influenza virus [52]. The probiotics, mainly those of the *lactobacillus* and *bifidobaterium* genera, such as *Lactobacillus plantarum* [53,54], *Lactobacillus rhamnosus* GG [55], *Lactobacillus paracasei* [23], *Bifdobacterium bifdum* [56], and *Bifidobacterium animalis* [16], can combat the influenza virus infection through regulating the immune response. The probiotic of *Pediococcus pentosaceus* has been shown to have immunomodulatory effects [30,31,32]. However, it is not known if it has an anti-influenza effect. Herein, we demonstrated that the *Pediococcus pentosaceus* MIANGUAN2 isolated from the feces of healthy individuals could alleviate influenza virus infection. Influenza infection can cause influenza pneumonia, a severe complication association with high viral load and a severe inflammatory cytokine response in the lung [57,58,59]. Thus, therapeutics targeting inflammation and virus load should be an ideal strategy for treating influenza infection [60]. Previous studies have demonstrated that treatment with certain probiotics can reduce influenza viral titer and proinflammatory cytokine levels in the lung. For example, oral administration of *Lactobacillus plantarum* GUANKE [53] or *Lactobacillus paracasei* MI29 [23] can inhibit influenza virus replication in the lung tissue of mice. Treatment with *Bifidobacterium pseudolongum* [16] or *Bacteroides* dorei [15] in mice significantly reduced the levels of proinflammatory cytokines in the lung. Similarly, in this study, we found that oral administration of MIANGUAN2 significantly decreased the viral load and levels of proinflammatory cytokines, including TNF-α, IL-1β, IL-12p70, and IFN-γ in the lung. The influenza virus triggers inflammatory cytokine production via activating inflammatory signaling pathways, such as the apoptosis, TNF, and NF-Kappa B signaling pathways [61,62]. Reducing the levels of inflammatory cytokines through inhibiting the expression of inflammatory pathways is critical for fighting influenza infection [63]. Notably, in the study, we observed that treatment with MIANGUAN2 decreased the expression of apoptosis, TNF, and NF-Kappa B signaling pathways, which further supports the anti-influenza effect of MIANGUAN2. Influenza virus infection also can be accompanied by intestinal symptoms, including intestinal inflammation, immune injury, and colon shortening [10]. A previous study has shown that treatment with *Bacteroides* dorei can restore the colon length of influenza infection mice to a certain extent [15]. Similarly, in our study, we found that oral administration of MIANGUAN2 also increased the length of the colon in an influenza infection mouse model. As such, MIANGUAN2 has excellent efficacy in terms of anti-influenza infection.

A large amount of research proves that the changes in gut microbiota play a significant role in the pathogenesis of influenza. Destroying gut microbiota by antibiotics can cause severe influenza symptoms [11]. In addition, influenza infection can cause the loss of food intake. Food restriction influences the abundance of bacteria using the fermentation product of dietary fibers and complex carbohydrates, as well as other gut commensals relying on the host mucins as an energy source [64]. The healthy gut microbiota can be disrupted after infection with the influenza virus, due to the loss of food intake and intestinal inflammation response [7,8,9,10], leading to worsening outcomes of influenza infection. Oral administration of probiotics is an appropriate method to regulate gut microbiota, which plays a significant role in preventing influenza infection. Song et al. reported that oral administration of *Bacteroides* dorei decreased the harmful bacteria *Escherichia* and *Shigella* and increased the beneficial bacteria *Lactobacillus*, *Prevotella,* and *Bacteroides* in influenza-infected mice [15]. In agreement, we found that treatment with MIANGUAN2 reduced the abundance of the harmful bacteria *Escherichia_Shigella* and *Enterobacter*. In addition, a previous study found that oral administration of *Lactiplantibacillus plantarum* 0111 effectively restored the intestinal imbalance caused by H9N2 infection [54]. In our study, we observed that influenza infection decreased the diversity of the gut microbial community (Shannon index), while oral administration of MIANGUAN2 restored the diversity of gut microbiota. Furthermore, it is worth noting that some beneficial bacterial communities, such as f_Lachnospiraceae and f_Ruminococcaceae, were enriched in the MIANGUAN2 group, and these bacterial strains are known to ferment non-digestible carbohydrates into SCFAs that are beneficial to the host [51]. *Akkermansia*, as a next-generation probiotic, has been generally recognized as being beneficial to host health [65]. However, we found that the abundance of *Akkermansia* significantly increased in the virus group compared with the control group, which is consistent with the results of previous studies [8,16]. Deriu et al. have found that influenza infection can increase the levels of Muc2 [66] (mucoprotein, the essential ingredient to maintain the growth of A. muciniphila), which may be the reason for the upregulation of *Akkermansia* in influenza infection mice. The lower relative abundance of *Akkermansia* in the MIANGUAN2 group may be due to the higher diversity of gut microbiota, which limits the growth of *Akkermansia*. Thus, although *Akkermansia* was enriched in the virus group, we cannot exclude its beneficial effect during influenza infection [67].

The species and concentration of microbial fermentation derived-metabolites are associated with the composition of gut microbial taxa [68]. Treatment with MIANGUAN2 can increase the abundance of SCFA-producing bacteria including p_Firmicutes, f_Lachnospiraceae, and f_Ruminococcaceae. Consistently, we observed the enrichment of some pathways associated with SCFA production in the MIANGUAN2 group through the metabolome analysis of cecal contents. The butanoate metabolism pathway and fatty acid degradation pathway are closely related to SCFA production [69]. Carbohydrates and undigested protein can translate to SCFAs through numerous biochemical pathways [70,71]. Lysine is one of the substrates to produce butyrate [72]. Moreover, after influenza infection, oral administration with MIANGUAN2 increased the levels of microbial fermentation-derived SCFAs including acetic acid, propionic acid, butyric acid, isobutyric acid, and valeric acid. Correlation analysis showed strong correlations between SCFAs and the gut microbiota that were enriched in the MIANGUAN2 group. Together, these results suggest that treatment with MIANGUAN2 may regulate the gut microbiota composition and increase the SCFAs levels in influenza-infected mice. 

The SCFAs are anti-inflammatory compounds with immunomodulatory properties [73], and they are produced in the gut and are able to reach systemic circulation to be transported to different organs [74]. SCFAs play an immunomodulatory role by binding to distinct receptors including the G protein-coupled receptors (GPCRs) GPR41 (FFAR3), GPR43 (FFAR2), and GPR109a (HCA2) that express in some immune cells or tissues [75], to affect the progress of the disease. Studies have demonstrated that gut microbial fermentation-derived SCFAs can improve distal pulmonary disease outcomes [75,76]. SCFAs have been detected in sputum, providing evidence of a connection between the gut and the lung [77]. According to previous studies, SCFAs have anti-influenza virus infection efficacy. Valentin and collaborators have demonstrated that treatment with acetate can enhance the killing activity of alveolar macrophages to fight against respiratory virus or bacteria infection in a free fatty acid receptor 2 (FFAR2)-dependent manner [8]. In addition, Trompette et al. have reported that treatment with high-fiber diets can alleviate influenza virus infection in mice. This treatment is associated with increased levels of SCFAs, especially butyrate, which boosted the effector function of CD8+ T cells [78]. A soluble dietary fiber named partially hydrolyzed guar gum (PHGG) can also increase the levels of SCFAs by regulating gut microbiota, resulting in inhibited inflammatory response to influenza infection [79]. Furthermore, Junling Niu and collaborators have shown that the acetate derived from *Bifidobacterium pseudopodium* NjM1 can protect against influenza virus infection through a GPR43-MAVS-IRF3-mediated IFN-I signaling [21]. In the present study, we found that gut microbial-derived SCFAs had a significantly negative correlation with the influenza infection phenotype and the proinflammatory factors, such as TNF-α, IL-1β, IL-12p70, and IFN-γ, indicating that MIANGUAN2 can also improve the influenza virus infection though increasing SCFA production to some extent. However, we cannot rule out that in addition to increasing the levels of SCFAs, MIANGUAN2 might fight against influenza infection through other mechanisms. Previous studies have shown that probiotics improve the response to influenza infection through several possible mechanisms, such as enhancing the innate and adaptive immunity of the host, blocking the entry of the virus into host cells, and the secretion of metabolites exhibiting antiviral properties [17]. Future studies are needed to further explore the other mechanisms of the anti-influenza activity of MIANGUAN2.

## 5. Conclusions

In summary, our study demonstrates that the novel *Pediococcus pentosaceus* MIANGUAN2 isolated from the feces of healthy individuals has a beneficial effect in protecting against influenza infection in mice, as indicated by how it reduced the lung viral titer and inflammation and improved lung pathological status. To our knowledge, this is the first report that evaluates the effect of *Pediococcus pentosaceus* in protecting against influenza virus infection. Furthermore, treatment with MIANGUAN2 can regulate the gut microbiota composition and increase the levels of SCFAs in influenza infection mice, which may be the underlying mechanism for MIANGUAN2’s protective effect against influenza virus-induced lung infection and inflammation (Figure 6E). Collectively, our results suggest that oral administration of MIANGUAN2 is effective in protecting mice against influenza virus infection, and future clinical trials are warranted to test its application in humans. 

## Figures and Tables

**Figure 1 nutrients-16-01923-f001:**
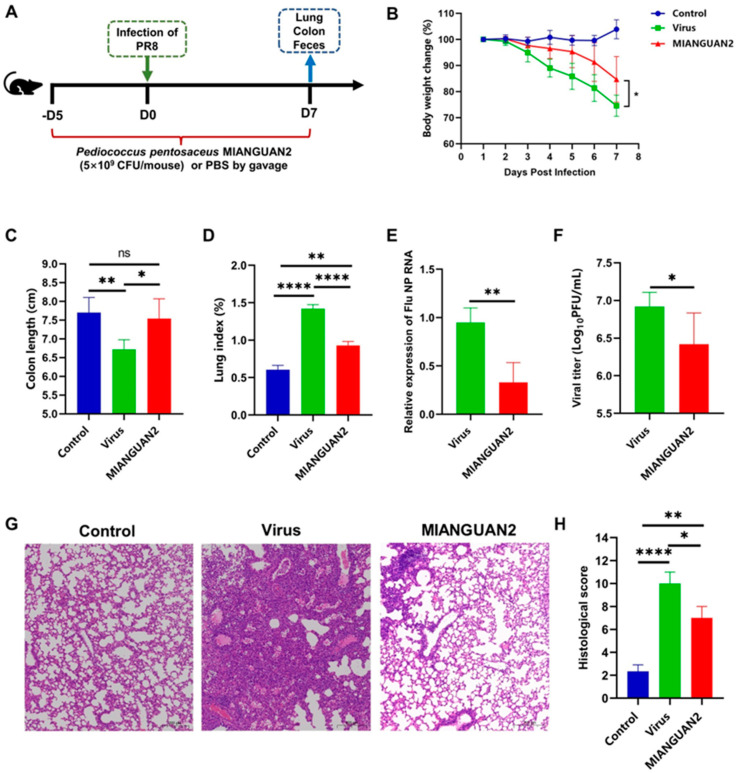
Oral administration of *Pediococcus pentosaceus* MIANGUAN2 protects the host against influenza infection. (**A**) The procedure of the animal experiments. Fifteen 13–15 g female C57BL/6J mice were randomly divided into three groups, as follows: the control group (non-infection), the virus group (influenza virus + PBS), and the MIANGUAN2 group (influenza virus + MIANGUAN2). The mice in infection groups were gavaged with 200 μL of PBS or MIANGUAN2 from day −5 to day 7, and the samples were collected at day 7 for further analysis. (**B**) The changes in mice body weight post-infection (p.i.). (**C**) The colon length of mice in the three groups at day 7 p.i. (**D**) The lung index of mice in the three groups at day 7 p.i. (**E**,**F**) The viral load in lung tissues at day 7 p.i. (**G**,**H**) Pathology and scores of lung tissues at day 7 p.i. Error bars represent means ± SD (**B**–**D**,**F**,**H**) or median with interquartile range (**E**). Significance was determined using two-tailed Student’s *t*-test (**B**,**F**), one-way ANOVA with a post hoc Tukey’s test (**C**,**D**,**H**), or a Mann-Whitney U test (**E**) based on the normality of data assessed by the Shapiro-Wilk normality test. * *p* < 0.05, ** *p* < 0.01, **** *p*  <  0.0001, and ns, not significant.

**Figure 2 nutrients-16-01923-f002:**
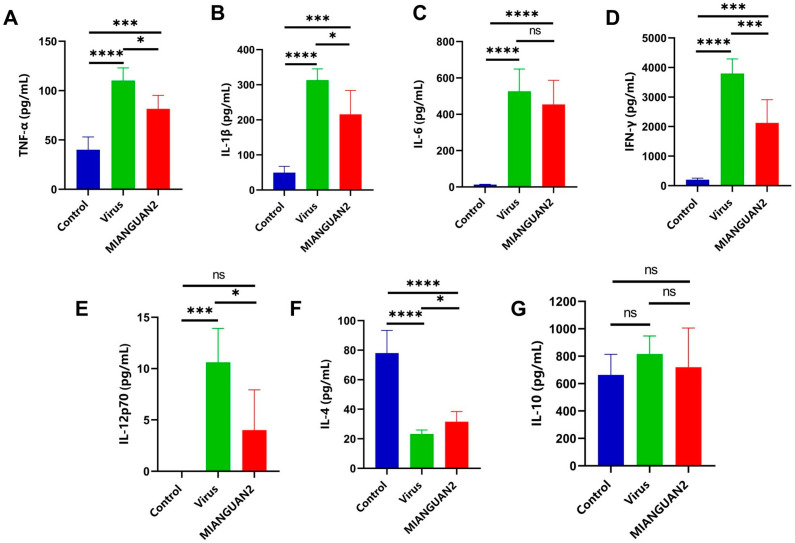
Oral administration of *Pediococcus pentosaceus* MIANGUAN2 alleviates lung inflammation. (**A**–**G**) The levels of proinflammatory and anti-inflammatory cytokines in the lung tissues of mice at day 7 p.i., shown as follows: (**A**) TNF-α, (**B**) IL-1β, (**C**) IL-6, (**D**) IFN-γ, (**E**) IL12p70, (**F**) IL-4, and (**G**) IL-10. All data are means ± SD. Groups were compared using one-way ANOVA with post hoc Tukey’s test based on the normality of data assessed by the Shapiro-Wilk normality test. * *p* < 0.05, *** *p* < 0.001, **** *p*  <  0.0001, and ns, not significant.

**Figure 3 nutrients-16-01923-f003:**
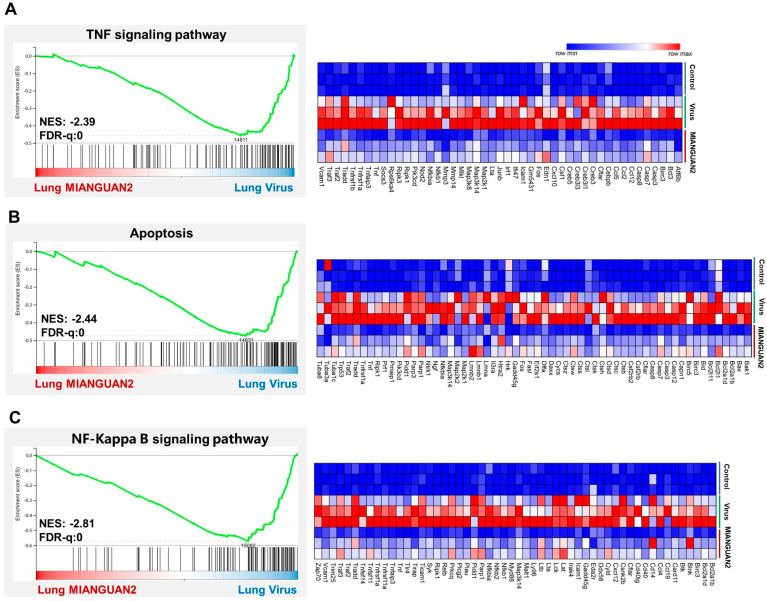
Oral administration of *Pediococcus pentosaceus* MIANGUAN2 can downregulate multiple inflammatory signaling pathways in the lung. (**A**−**C**) GSEA analysis showed the inhibition of TNF (**A**), apoptosis (**B**), and NF-Kappa B signaling pathways (**C**) in the lung tissues of the MIANGUAN2 group compared with the virus group. The clustering heatmaps on the right display the gene expression profiles with dominant roles in the three inflammation pathways, respectively. GSEA: gene set enrichment analysis, NES: normalized enrichment score, FDR-q: false discovery rate *q* value.

**Figure 4 nutrients-16-01923-f004:**
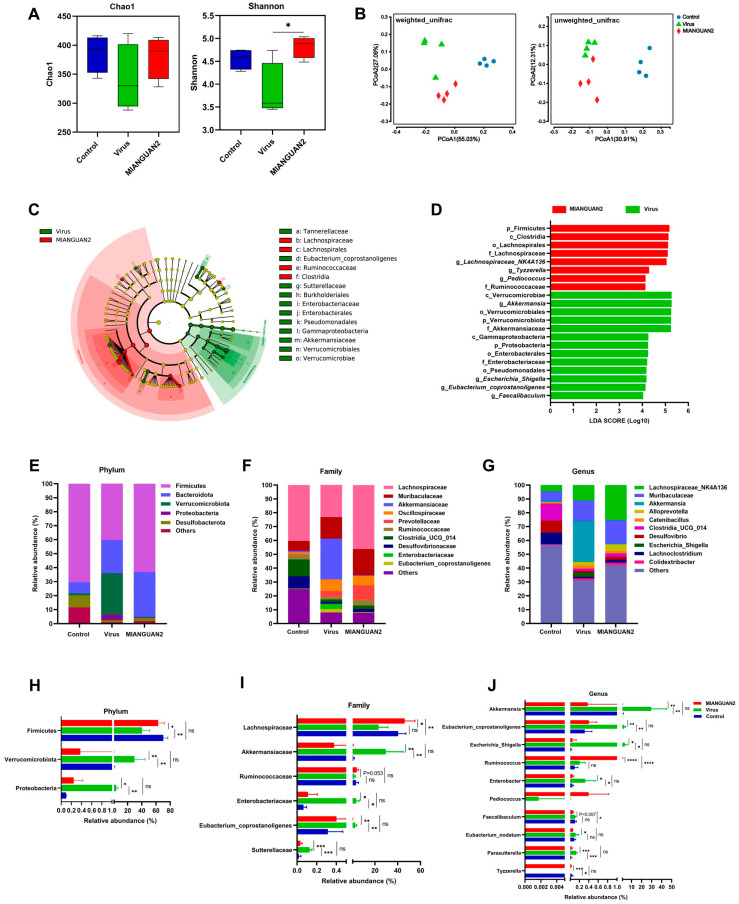
Oral administration of *Pediococcus pentosaceus* MIANGUAN2 alters the composition of gut microbiota in influenza-infected mice. (**A**) Boxplots of the Chao1 and Shannon indexes of alpha diversity index among the three groups. (**B**) PCoA plots based on weighted_unifrac metrics or unweighted_unifrac metrics. (**C**) The LEfSe taxonomic cladogram. (**D**) LDA score plot with a threshold of 4.0. (**E**–**G**) The top 5 most abundant microbial phyla, as well as the top 10 most abundant bacterial families and genera, are displayed among the 3 groups via stacked bar plots, respectively. (**H**–**J**) Bar graphs display the relative abundances of differential bacterial taxa at the phylum, family, and genus levels, respectively. Error bars represent Min to Max (**A**) or means ± SD (**H**–**J**). Groups were compared using the Kruskal-Wallis test with Dunnett’s multiple comparison test (**A**: Shannon) or ordinary one-way ANOVA with a post hoc Tukey’s test (**H**–**J**) based on the normality of data assessed by the Shapiro-Wilk normality test. * *p* < 0.05, ** *p* < 0.01, *** *p* < 0.001, **** *p*  <  0.0001, and ns, not significant.

**Figure 5 nutrients-16-01923-f005:**
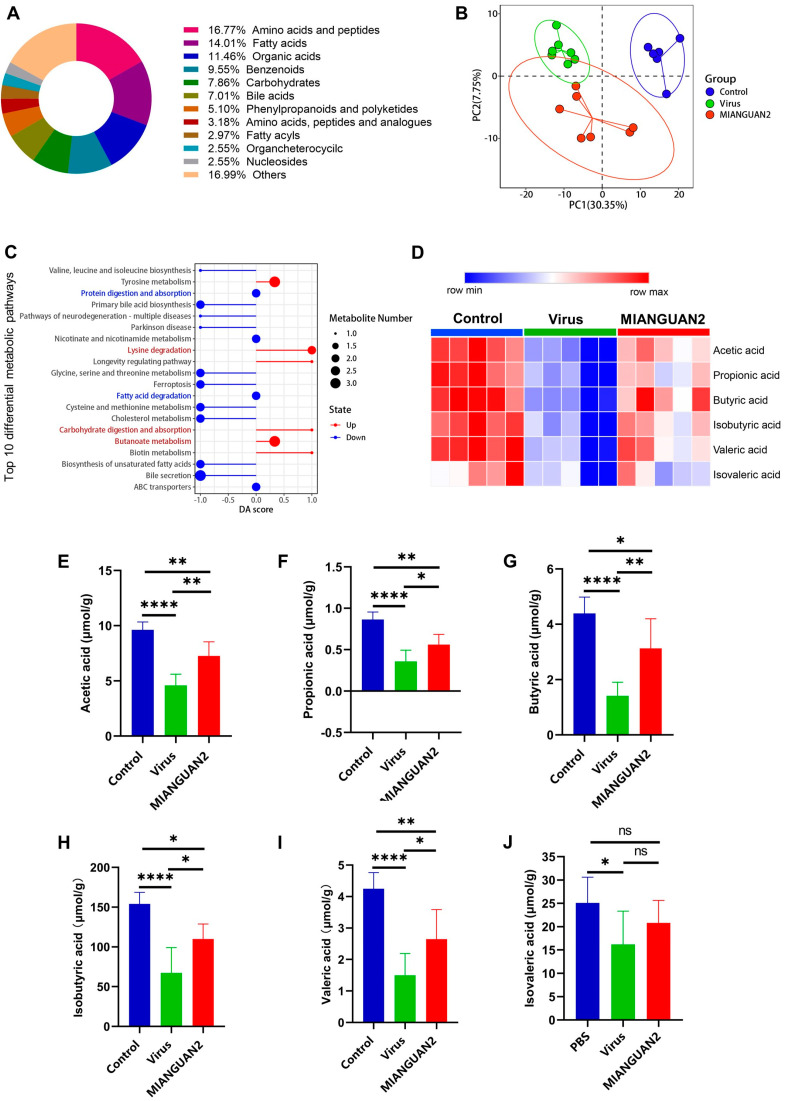
Oral administration of *Pediococcus pentosaceus* MIANGUAN2 increases SCFA levels in the cecal content of influenza-infected mice. (**A**) The metabolite classification ring diagram displays the top 10 most abundant metabolites according to the classification of the final class. (**B**) The partial least squares-discriminant analysis (PLS-DA) plot among the three groups. (**C**) The bubble plot of differential abundance score (DA score) between the MIANGUAN2 group and the virus group. (**D**) The clustering heatmap depicts the SCFAs profiles in mice cecal content among the three groups. (**E**–**J**) The comparison of SCFAs concentration in cecal content of the three groups, including (**E**) acetic acid, (**F**) propionic acid, (**G**) butyric acid, (**H**) isobutyric acid, (**I**) valeric acid, and (**J**) isovaleric acid. Data in (**E**–**J**) are means ± SD and were analyzed using one-way ANOVA with post hoc Tukey’s test based on the normality of data assessed by the Shapiro-Wilk normality test. * *p* < 0.05, ** *p* < 0.01, **** *p*  <  0.0001, and ns, not significant.

**Figure 6 nutrients-16-01923-f006:**
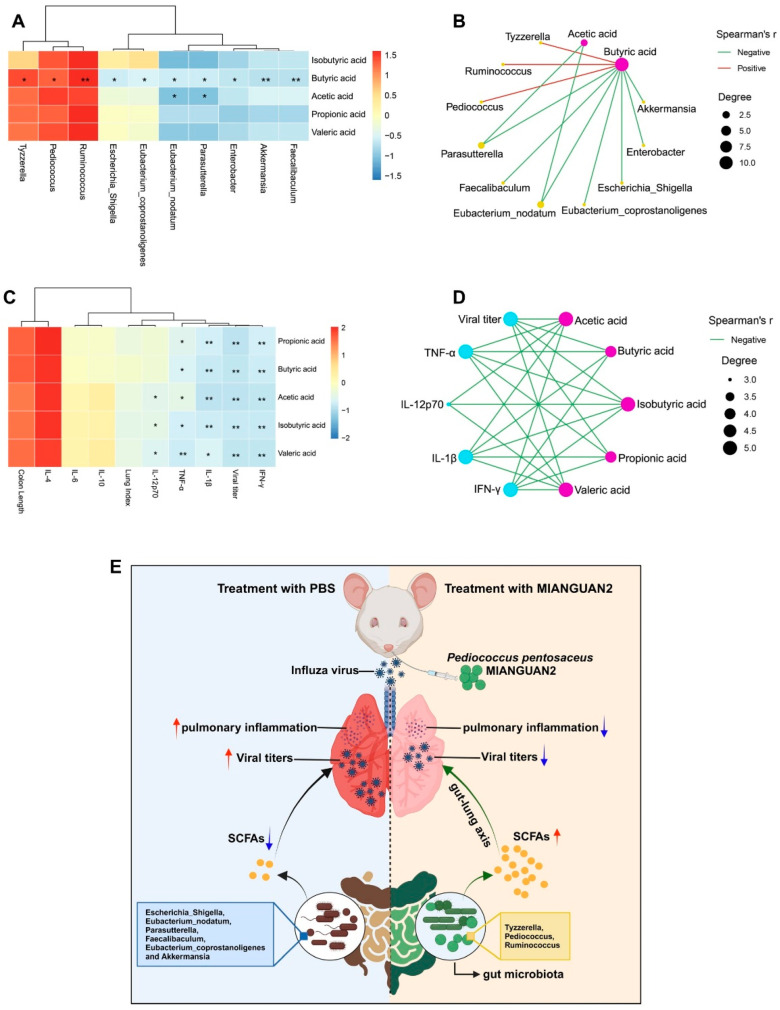
The Spearman’s correlation analysis of the gut microbial derived-SCFAs vs. gut microbiota or influenza infection phenotype. The heatmaps show the Spearman correlation coefficients between the gut microbial fermentation derived-SCFAs and (**A**) relative abundances of differential bacterial genera or (**C**) the influenza infection phenotype. The network diagrams display the significant correlation (*p* < 0.05) between the gut microbial fermentation derived-SCFAs and (**B**) relative abundances of differential bacterial genera or (**D**) the influenza infection phenotype. (**E**) The potential mechanism underlying the effects of MIANGUAN2 on influenza virus infection. Significant differences are denoted by * *p* < 0.05 and ** *p* < 0.01.

## Data Availability

All sequencing data are publicly available in the NCBI SRA database (BioProject numbers: PRJNA1109985 and PRJNA1110029).

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
