# Peer review of "Pediococcus pentosaceus MIANGUAN2 Alleviates Influenza Virus Infection by Modulating Gut Microbiota and Enhancing Short-Chain Fatty Acid Production"

_nutrients, 2024, doi:10.3390/nu16121923_

Round 1
Reviewer 1 Report
Comments and Suggestions for Authors
The aim of this study was to evaluate the effects of P. pentosaceus MIANGUAN2 administration on influenza infection in mice.show
The manuscript is well written and the results are interesting and display a certain degree of novelty. However, the statistical analysis and some other aspects should be improved before publication.
General commentaries
· The names of bacterial genera and species must be italicized throughout the manuscript.
· The authors frequently use the term “symptoms”, which generally refers to clinical symptoms. However, the study does not report any clinical symptoms related to influenza in the animals, and therefore this term should be deleted in the title and elsewhere in the text.
· All figures in the manuscript should indicate whether the results obtained in the MIANGUAN2 group are significantly different from those in the control group. This would help to determine whether the restoration observed with the probiotic for each variable is total or partial.
Specific commentaries
Introduction
· L62: B. animalis and not B. animals
· The authors have made several statements, L49-51, L53, which should be deepened by describing the possible mechanisms involved in the results described.
· L37-71: Please specify from which substrates GlcNAc and DAT are produced by these bacteria
· L75-77: How does 18-HEPE promote resistance to viral infection?
· L78-79: explain this sentence better.
Methods
· L109: Explain why female mice rather than males were used in this study.
· L125: Explain what the lung index consists of.
· L217-28 The statistics used to compare the results seem inappropriate. Results should be expressed as mean ± SEM if parametric statistics are used, and as median and interquartile range if non-parametric are used. Indicate how the normal distribution of the data was tested. In all figures, results are presented as means and SEM, suggesting that the data distribution is normal when it is likely that many of them are not. The type of analysis performed (ANOVA or Kruskal-Wallis) must be indicated in each figure.
Results
· In Figure 1c, the authors show a decrease in colon length in the infected animals. They should explain whether similar results have been described in other studies and what are the mechanisms explaining such findings.
· L139: there are not 3 independent experiments. Please correct.
· Figure 2-H is very small and it is difficult to see the text. It would be probably better to present it as an independent figure. Please explain what GSEA means in L261 and K269.
· Figures 4-E to J: According to the units and scales used in these figures, the more concentrated SCFAS were isobutyrate and isovalerate, while the less concentrated were propionate, butyrate and valerate, with acetate 15 times less than isobutyrate!!! These results do not correspond to the those described in the literature, where acetate is the most concentrated, followed by propionate and butyrate, then valerate, with isobutyrate and isovalerate present only in low concentrations. Therefore, these results/figures need to be revised.
· L365-76 and figure 5: these results show correlations between SCFA levels and bacterial and inflammatory markers, not symptoms. Please correct in the text and figure legend.
Discussion
· Please describe the mechanisms by which influenza infection may affect the composition and diversity of the gut microbiota.
· L425: Change “complicated carbohydrates” to “non-digestible carbohydrates”
· L429: The sentence “… although the amount of Akkermansia is similar in all groups” is unclear as the study did not determine the absolute numbers of bacteria. Please explain. In addition, the authors should propose a mechanism by which influenza could increase Akkermansia. Discuss the fact that this genus is generally considered to be protective in the digestive ecosystem.
· L447: Can the SCFAs produced in the colon reach the lungs? Are there SCFA receptors in lung tissue?
Comments on the Quality of English Language
Minor editing of English language required
Reviewer 2 Report
Comments and Suggestions for Authors
This study is considered well-designed animal research with well-executed presentation and interpretation of the results. The strain used by the authors in this study is a newly isolated strain, and they conducted research to demonstrate its immune-enhancing effects as a probiotic. However, it is recommended to add a review comparing the immune-enhancing effects of other strains. Does this probiotic have a greater impact on the intestinal environment compared to other strains? This explanation also needs to be supplemented in the manuscript. Are there any other mechanisms besides immune cell regulation by short-chain fatty acids? Short-chain fatty acids could be a result of probiotic intake, and there is also the possibility of immune regulation effects through other mechanisms. These points should be added to the discussion.
Reviewer 3 Report
Comments and Suggestions for Authors
Pediococcus pentosaceus is gradually attracting attention, leading to a rapid increase in experimental research. In addition, various strains of P. pentosaceus have been highlighted for probiotic use due to their anti-inflammation, anticancer, antioxidant, detoxification, and lipid-lowering abilities. It has been known from the literature that gut microbiota modulates the host response to protect against influenza infection, but mechanistic details remain largely unknown.
The authors described the possible mechanism by which these beneficial effects could happen.
My question is whether, in the present experimental research from the group of Pediococcus pentosaceus MIANGUAN2, all five animals had improved clinical symptoms from severe to mild, or was total recovery observed in any of them? What was the number of animals with clinical improvement?
Moreover, what was the percentage of reduced levels of inflammatory cytokines in the group of Pediococcus pentosaceus MIANGUAN2 versus the control group?
The methodology is appropriate. The manuscript is well written, and the discussion/conclusions are acceptable.
Overall, the data is of interest to future research.
Comments on the Quality of English Languagenone
